# Innovative Therapy Combining Neck Muscle Vibration and Transcranial Direct Current Stimulation in Association with Conventional Rehabilitation in Left Unilateral Spatial Neglect Patients: HEMISTIM Protocol for a Randomized Controlled Trial

**DOI:** 10.3390/brainsci13040678

**Published:** 2023-04-18

**Authors:** Sarah Millot, Jean-Marie Beis, Jonathan Pierret, Marina Badin, Verginia Sabau, Laurent Bensoussan, Jean Paysant, Hadrien Ceyte

**Affiliations:** 1UGECAM Nord-Est, Institut Régional de Médecine Physique et de Réadaptation, Centre de Médecine Physique et de Réadaptation, Lay Saint-Christophe, France; 2Université de Lorraine, DevAH, Nancy, France; 3UGECAM PACA-Corse, Centre Helio Marin, Vallauris, France; 4Aix Marseille Univ, CNRS, INT, Marseille, France; 5UGECAM PACA, Institut Universitaire de Réadaptation de Valmante Sud, Marseille, France; 6Aix Marseille Univ, CNRS, ISM, Marseille, France

**Keywords:** neglect, spatial perception, neck muscle vibration, transcranial direct current stimulation, occupational therapy sessions

## Abstract

Unilateral spatial neglect (USN) rehabilitation requires the development of new methods that can be easily integrated into conventional practice. The aim of the HEMISTIM protocol is to assess immediate and long-term recovery induced by an innovative association of left-side neck-muscle vibration (NMV) and anodal transcranial Direct Current Stimulation (tDCS) on the ipsilesional posterior parietal cortex during occupational therapy sessions in patients with left USN. Participants will be randomly assigned to four groups: control, Left-NMV, Left-NMV + sham-tDCS or Left-NMV + anodal-tDCS. NMV and tDCS will be applied during the first 15 min of occupational therapy sessions, three days a week for three weeks. USN will be assessed at baseline, just at the end of the first experimental session, after the first and third weeks of the protocol and three weeks after its ending. Our primary outcome will be the evolution of the functional Catherine Bergego Scale score. Secondary outcome measures include five tests that investigate different neuropsychological aspects of USN. Left NMV, by activating multisensory integration neuronal networks, might enhance effects obtained by conventional therapy since post-effects were shown when it was combined with upper limb movements. We expect to reinforce lasting intermodal recalibration through LTP-like plasticity induced by anodal tDCS.

## 1. Introduction

Unilateral spatial neglect (USN) is a perplexing neuropsychological syndrome that affects approximately 30% of stroke-affected individuals worldwide [1]. This syndrome results from hemispheric lesions in cortical structures (posterior parietal cortex, superior temporal gyrus, temporo-parietal junction, and less frequently frontal cortices), subcortical structures (thalamus, basal ganglia) or white matter fiber tracts (fronto-parietal pathways) [2,3], with a greater prevalence after right brain lesions rather than left ones (38% and 18%, respectively [1]). These brain networks are known to be involved in the construction of the spatial coordinate system of egocentric and allocentric references frames [4]. Therefore, damage to these networks can lead to USN in all space sectors: extra-personal, peri-personal and personal spaces. In concrete terms, this syndrome affects spatial awareness and is characterized by the inability to detect or respond to stimuli from the contralesional side of space without any dysfunction in the capture or transmission of sensory information to the central nervous system. These spatial disorders have disabling functional consequences that can be observed during activities of daily living such as reading, dressing, shaving, eating or getting from one place to another [5].

It is well established that USN is an independent factor associated with poor prognosis and functional outcome after stroke [6], and consequently, the rehabilitation of this syndrome is a challenging issue. Conventional rehabilitation of USN is mostly based on visual scanning training, the first developed method, which aims at better visual exploration toward the contralesional side [7]. The effectiveness of this “top–down” approach is based on the stimulation of attentional resources and requires patients to be aware of their impairment in order to actively correct it. However, since attentional capacities are often altered after stroke and especially in patients affected by USN [8], other rehabilitation methods, called “bottom–up” approaches, favor sensory manipulations either by deceiving the patient’s body perception (vestibular stimulation, optokinetic stimulation, neck muscle vibration), which is costless in terms of attentional resources, or through cognitive adaptation induced by visuo-haptic distortion (visual prisms) [9]. Both showed interesting results in early-phase rehabilitation [10,11]. Among these “bottom–up” techniques, neck muscle vibration (NMV) consists in applying a 80–100 Hz vibration on the skin over the upper fibers of one of the trapezius muscles, creating a proprioceptive illusion such that the body midline and surrounding space are perceived to be shifted to the stimulated side [12]. Some studies have shown a persistent reduction in USN symptoms thanks to left-side NVM [13,14]. For instance, Kamada et al. [15] showed that applying this illusory technique for 5 min right before an occupational therapy session optimized its beneficial effects as quantified by the conventional Behavioral Inattention Test (c-BIT) [16], a neglect-specific test battery, and the Functional Independence Measure (FIM) [17]. In addition, several studies suggest that combining different therapeutic approaches with converging effects may lead to the best benefits on USN [18,19]. Left-side NMV appears to be ideal for combination with other rehabilitation techniques, given its safety, simplicity of use, passive nature and particular therapeutic effects. USN reduction was shown to persist when NMV was coupled with visual scanning training [20] or applied in a visuo-haptic feedback context [21]. This feedback context was achieved through repeated reaching movements similar to the motor activities usually performed during patients’ conventional rehabilitation sessions. These results therefore support the interest of associating the left-side NMV technique simultaneously to rehabilitation sessions for left USN patients.

A third category of treatment, that can neither be classified as “top–down” nor “bottom–up”, based on non-invasive brain stimulation such as repetitive transcranial magnetic stimulation (rTMS) or transcranial direct current stimulation (tDCS), is used to modulate neuroplasticity by increasing (low-frequency rTMS or anodal tDCS) or decreasing (high-frequency rTMS or cathodal tDCS) cortical excitability [22]. Recent meta-analyses on USN showed a significant benefit of non-invasive brain stimulations when cortical excitability is increased in the ipsilesional hemisphere and/or decreased in the contralesional hemisphere [23,24]. Many studies using different modes and cortical localizations of tDCS, based on a low-amplitude electrical current, have shown its effectiveness in reducing motor and cognitive disorders after a stroke [25]. Conventionally, the two main modes of tDCS (anodal and cathodal) imply that the stimulation electrode (anode for anodal tDCS and cathode for cathodal tDCS, respectively, a-tDCS and c-tDCS) is positioned on the cortical area to be modulated, the reference electrode being placed remotely (for example on the supra-orbital area) [22]. For instance, a single a-tDCS session on the ipsilesional posterior parietal cortex (i-PPC) [26] or c-tDCS on the contralesional posterior parietal cortex (c-PPC) [27] was sufficient to induce a significant improvement on neuropsychological performances in left USN patients. Beneficial effects on USN lasting up to one week were obtained after applying a-tDCS or c-tDCS (respectively, on the i-PPC or the c-PPC) for three weeks during daily occupational therapy sessions [28]. A third mode of tDCS, referred to as dual or bi-hemispheric, involves placing the two electrodes on the cortex on either side of the midline, aiming at modulating simultaneously and oppositely the excitability of the two hemispheres [29]. This dual mode in a bi-parietal configuration has also been successful in reducing USN symptoms [30,31]. Other studies recently evaluated the efficacy of tDCS in association with “bottom–up” rehabilitation techniques (visual prisms [32] and optokinetic drift [33]). Làdavas et al. [32] showed that the well-established efficacy of prism adaptation for reducing USN symptoms [11], applied in this study by daily 30-min sessions for two weeks, was significantly magnified by a 20-min application of a-tDCS on the i-PPC before (10 min) and partially while wearing visual prisms (10 first minutes). Similarly, Turgut et al. [33] showed significant benefits lasting up to 6 days after applying a therapeutic protocol combining dual-tDCS (anode on the i-PPC and cathode on the c-PPC) and optokinetic drift for 20 min eight times over two weeks.

USN rehabilitation remains a public health issue in terms of patients’ independence and healthcare costs. One of its main challenges lies in offering therapeutic innovations that can be easily integrated into conventional practice [34]. NMV and tDCS appear to be easy-to-use techniques that could meet this core condition for the rehabilitation of USN. The aim of the present protocol study is to assess beneficial effects on USN, specifically on immediate and long-term functional outcomes and neuropsychological aspects of spatial recovery, induced by a combined treatment associating the left-side NMV technique and a-tDCS on the i-PPC during occupational therapy sessions in patients with left USN. Our hypothesis is that the increase in excitability of the lesioned hemisphere induced by a-tDCS will amplify the lasting beneficial effects of left-side NMV during conventional occupational therapy sessions.

## 2. Materials and Methods

This multicentric prospective randomized controlled superiority trial will focus on assessing the benefits of our therapeutic protocol on independence in daily life activities, neuropsychological aspects of different spatial components of USN (personal, extra-personal and representational neglect) and egocentric spatial perception assessments, using a mixed measurement design. The HEMISTIM protocol will take place in three neurological rehabilitation centers in Grand-Est and Provence-Alpes Côte d’Azur regions, France.

### 2.1. Ethical Consideration and Trial Registration

#### 2.1.1. Informed Consent

Our clinical trial will be conducted in accordance with the Declaration of Helsinki. Oral and signed informed consent will be requested by an investigator after patients have received oral and written information related to the study. Participants can revoke their consent at any time.

#### 2.1.2. Data Management

The data will be recorded in an observation book by the evaluator in charge of the patient, written in a case report form and stored according to French legislation. A data monitoring committee will not be necessary in this context of intervention with minimal risk and constraints and without major modification of patients’ usual care. The confidentiality of collected data will be ensured by pseudonymization. The final dataset will be owned by the trial sponsor and accessible to all investigators, according to French legislation.

#### 2.1.3. Trial Registration

This study protocol was registered on the clinicaltrial.gov registry: NCT05281302. The manuscript is in accordance with the SPIRIT guidelines (http://www.spirit-statement.org/) for reporting study protocols. It has already been validated by the scientific and ethics committee of Institut Régional de Médecine Physique et de Réadaptation of Nancy, and an application is currently underway with an independent national ethics committee under the number 2022-A01673-40.

### 2.2. Eligibility Criteria

Patients aged at least 18 years old and admitted in a post-stroke rehabilitation unit in the sub-acute phase (15 days to 6 months after the onset of stroke symptoms) will be eligible for the study. Patients with left USN subsequent to a first unilateral right hemispheric stroke will be included. The diagnosis of stroke will have to be confirmed by computed tomography or magnetic resonance imaging. Screening for USN will be performed by the c-BIT [16], a score ≤ 129 defining USN. Patients unable to give informed consent, pregnant women, patients with skin lesions on the areas for electrode placement or having history of metal-in-cranium injury, epilepsy, vestibulo-cochlear illness or cardiac pacemaker will not be included.

### 2.3. Materials

Vibratory stimulators (VibramoovPhysio^®^, Techno Concept, Manosque, France) will be fixed bilaterally on the skin over the belly of the trapezius muscles and fastened with straps. Only the left-side NMV vibrator will be activated at a frequency of 100 Hz and an amplitude of 300 μm for 15 min, alternating 150 s of stimulation and 30 s of pause five times, as in previous studies [21,35].

tDCS will be applied using an electrical stimulator (DC-STIMULATOR^®^, NeuroCare, Illmenau, Germany) through large saline-soaked sponge surface electrodes placed as follows: anode over the (right) i-PPC (P4 according to the International 10–20 EEG electrode placement system) and cathode over the left supraorbital region. Stimulation parameters were chosen according to Làdavas et al.’s study [32] (anodal, 2 mA amplitude for 20 min).

### 2.4. Design of the HEMISTIM Protocol

This study will take place during 7 weeks of a conventional 5-day rehabilitation (Monday to Friday) for each patient included in this study, as illustrated in Figure 1.

#### 2.4.1. Inclusion in Groups

The first week will allow us to include patients with left USN and to randomly assign them to four independent parallel groups (centralized randomization in a 1:1 ratio performed using a custom made algorithm on Matlab (MathWorks, Inc., Natick, MA, USA) by a blinded investigator: a control group receiving no additional treatment to conventional occupational therapy sessions and three experimental groups receiving in addition to the usual treatment either left-NMV, left-NMV + sham-tDCS or left-NMV + a-tDCS experimental treatments, administered by trained occupational therapists.

In these three experimental groups, each patient will be equipped with vibratory stimulators during conventional occupational therapy sessions taking place on Monday, Wednesday and Friday for three weeks from the second week on. The left-side NMV vibrator will be activated during the first 15 min of the occupational therapy session (lasting 45 min), as previously described [21,35]. In the two experimental groups receiving sham or a-tDCS, electrodes will be placed as described above. A continuous 2 mA current will be delivered for 20 min in the left-NMV + a-tDCS group, 5 min before (patients will be seated in a quiet room) and during the first 15 min of the occupational therapy session, simultaneously to the NMV stimulation. The current will be automatically turned off after 20 s in the left-NMV + sham-tDCS group, allowing the reproduction of the initial mild itching sensation at the beginning of active tDCS, thus ensuring that the patient stays blind to the activation status of the device. In the case of adverse events or unintended effects of trial interventions, the investigator who enrolled the patient will report its occurrence to the trial sponsor through a report form provided in the case report form.

#### 2.4.2. Setting of Interventions

During the first 15 min of the occupational therapy sessions, without modifying the classically performed exercises, the therapist will be invited to solicit voluntary upper limb movements in patients’ peri-personal space, without looking for performance or speed, but rather for a qualitative execution. The patients’ position will be verified to ensure that they are sitting in front of the table. In the initial experimental phase (during the first sessions), in order to maintain a maximum level of motivation, patients will have the option of mobilizing their ipsilesional upper limb instead of the contralesional one to perform these different exercises. The use of the latter will be gradually increased, avoiding as much as possible the patients performing bimanual activities in their neglected hemi-space [36,37].

No modification of the conventional rehabilitation will occur such as daily physiotherapy, speech therapy, adapted physical activities, Tuesday and Thursday occupational therapy sessions, and the second part of experimental occupational therapy sessions.

#### 2.4.3. Setting of Assessments

In this study, the primary outcome will be the evolution of the global USN behavior evaluated by the Catherine Bergego Scale (CBS) [38], a functional scale assessing USN severity in daily life situations.

In addition, various USN spatial cognitive-motor skills will be assessed as secondary outcomes by the following five tests.

The Fluff test [39] aims at investigating personal neglect with targets attached to patients’ clothes, as illustrated in Figure 2a. The number of removed targets will be counted in order to calculate a score of personal neglect expressed as the difference of percentages of targets found on each side of the body (left–right), this value becoming more negative with the severity of personal neglect.

Gainotti’s drawing copying test [40], illustrated in Figure 2b, informs us on two kinds of peri-personal neglect referring to distinct disturbances of mental representations involved: space-centered neglect (egocentric) and object-centered neglect (allocentric). Therapists will verify throughout the test that the patient remains centered in relation to the paper. The quality of the drawing will be evaluated with respect to allocentric and egocentric reference frames.

The Map of France test [41] is designed to investigate representational neglect. The number of cities mentioned by the patient will be counted on either side of a median line drawn vertically through the center of France (Lille–Perpignan axis; see Figure 2c). A mental evocation score will be calculated as the ratio of the number of cities on each side of this axis (left/right). In addition, the strategy and order of mention will be recorded for qualitative analysis.

The Subjective Straight-Ahead test (SSA) [21] investigates the accuracy of egocentric perception, using the experimental device pictured in Figure 2d. The centered position of the patient will be verified throughout the test. Egocentric perception will be expressed as the mean error of 20 successive trials, after measuring for each trial the angular difference between the measured angles of the hand pointed toward the SSA direction and the objective straight-ahead direction of the patient’s body. A higher score indicates a less accurate egocentric perception; leftward errors will be assigned as negative values and rightward errors as positive values.

The Wheelchair navigation test [21], illustrated in Figure 3, aims at investigating spatial navigation skills and extra-personal neglect. The ride will be carried out once in each direction (A-B and B-A) to balance the number of obstacles and turns to the left and right. The time taken to finish the course will be measured and the number of bumps counted. Navigation quality will also be observed qualitatively (obstacle avoidance, goal strategy, perseverations).

Each test will be carried out several times over the study by a trained occupational therapist:

Baseline assessment (Pre): During the first week, each test (except for the CBS realized only once) will be repeated three times in order to obtain a reliable estimation of the participants’ performance level (mean values) before starting the experimental interventions.

Follow-up assessments: These will take place just at the end of the first experimental session (Postsession1), after the first and third weeks of the protocol (Postweek1 and Postweek3, respectively) and there will be a final evaluation three weeks after its ending (Postweek6). In addition, all patients will be asked about their expectations and satisfaction regarding the treatment at the end of each week of intervention.

#### 2.4.4. Monitoring and Blinding

The trial sponsor will assign a supervisor to monitor 10% of the experimental occupational therapy sessions to ensure that they are conducted in accordance with the protocol. In addition, 10% of the evaluations will be monitored. The researcher who will generate the patients’ allocation sequence will not perform the neuropsychological evaluation, nor apply therapeutic stimulations. The therapist who will perform USN evaluations before and after the treatment will be blind to the patients’ group assignment.

### 2.5. Sample Size Calculation

To determine the number of left USN patients to include, we sought to perform an a priori headcount calculation based on published work in the field. However, none of them have combined NMV and tDCS or used the CBS score as the primary endpoint. No a priori power calculation could therefore be performed, but we were inspired by the number of patients included in the works of Làdavas et al. [32] and Turgut et al. [33], both of whom used tDCS on USN patients, the former combined with visual prisms and the latter with optokinetic drift. These two studies included 30 and 32 patients, respectively, which were divided for the first in two groups of 11 patients and one group of 8, and for the second in two groups of 16 patients. Thus, we estimated that 48 patients would be necessary for our study, randomly and homogeneously distributed in each of the 4 groups. To ensure that the target number of patients is recruited, three neurological rehabilitation centers admitting each year a large number of patients meeting the inclusion criteria will be involved. We will verify a posteriori that the groups do not differ in sex, mean age, education, duration of illness, or baseline CBS scores, and that they have similar stroke locations.

### 2.6. Statistical Analysis

CBS scores, those of personal neglect, mean egocentric errors and mental evocation, the time needed to finish the wheelchair course and the number of bumps in each direction will be entered as quantitative data.

After having verified the required assumptions about data distributions (Shapiro–Wilk test of normality, Levene’s test of homoscedasticity, Box’s M-test of covariance homogeneity and Mauchly’s test of sphericity), a mixed analysis of variance (with repeated measures for the SSA) will be performed on each quantitative variable with test periods (Pre, Postsession1, Postweek1, Postweek3, Postweek6) as within-subject factors and treatment groups (control, left-NMV, left-NMV + sham-tDCS, left-NMV + a-tDCS) as between-subject factors. Post hoc tests will be conducted using Tukey’s honestly significant difference method when needed, and *p*-values will be adjusted with the Bonferroni method. Size effects will be reported using the η2 (Eta-square) for each ANOVA and Cohen’s d for each Tukey’s post hoc test, along with confidence intervals.

If the conditions for using parametric tests are not met, then non-parametric statistical analyses of comparisons between test periods and groups will be performed. For each treatment group, we will perform a Friedman test to analyze the effect of the test period on each quantitative variable independently. When a Friedman test is significant, post hoc Wilcoxon signed rank tests with the Bonferroni adjustment method will be used to perform pairwise comparisons between each of the five time points. Kruskal–Wallis tests will be performed to compare the treatment groups. First, a Kruskal–Wallis test will be performed on the sum of ranks of the scores in each group before the first session in order to ensure that the initial mean scores (Pre) are equivalent between groups. Second, the effect of treatment will be calculated for each participant at each post-session assessment (Postsession1, Postweek1, Postweek3, Postweek6) by subtracting the mean initial score (Pre) from each quantitative data, independently. In order to analyze the effect of the treatment received on the evolution of quantitative data, Kruskal–Wallis tests will be performed on the sum of ranks of the Post–Pre score differences in each group. When the Kruskal–Wallis test is significant, post hoc pairwise comparisons between groups will be calculated, and the *p*-value will be adjusted by the Steel–Dwass–Critchlow–Fligner method.

Any participant with a missing score on any of the assessment sessions will be excluded from the statistical analysis for that specific test.

All statistical analyses will be performed using R software (version 2021.09.1 Build 372). The threshold for statistical significance will be set at α = 0.05.

### 2.7. Dissemination Policy

Patients will have the option of knowing their personal results and/or the results of the study after publication. The anonymous results of this research will be disseminated in internal communications, scientific conferences and published in scientific journal articles.

## 3. Discussion

The rehabilitation of spatial cognition disorders, such as the USN syndrome, is still a real challenge, which is partly due to the associated anosognosia and attentional disorders [8]. The HEMISTIM protocol may represent a partial solution by proposing an innovative therapeutic approach based on well-identified tools accessible to therapeutic teams. The aim of this protocol is to optimize conventional USN rehabilitation practices through the contribution of new scientific knowledge in the fields of neuromodulation and spatial cognition and more particularly the interest of augmented sensory feedback in spatial perception. In this respect, our previous translational research studies have addressed this question and identified the optimal sensori-motor context for the persistence of left-side NMV effects [35] and its benefits in a USN case study [21].

The perceptual effects of NMV are attributed to a non-specific activation of the contralateral hemisphere and the activation of vestibular neurons in the parieto-insular cortex [42]. These neurons with bilateral receptive fields participate in multisensory integration processes and internal representations of the position of the head and body in space [42]. Left-side NMV seems to counterbalance the egocentric representation deviation observed in USN patients through these central mechanisms [43]. The key question of persistent effects was addressed by Ceyte et al. [35], who showed that lasting post-effects were only present when left-side NMV was associated with upper limb self-activation, and these results were consistent with our previous case study on a USN patient [21]. Transferring our protocol directly into conventional rehabilitation sessions could further increase their benefits. Indeed, reaching toward a target, as it was performed in a visuo-haptic feedback context in our previous studies [21,35], is very similar to the motor activities performed during ecological upper limb rehabilitation sessions. Based on these findings, we assume that using this technique during occupational therapy sessions might enhance beneficial effects obtained by conventional rehabilitation without needing to add specific sessions as it is sometimes suggested. In addition, the positive impact expected from this initial protocol could be optimized by tDCS neuromodulation. Indeed, non-invasive brain stimulations are known to modulate cortical excitability and therefore seem an ideal tool to promote long-lasting neuroplasticity [44]. The duration, amplitude and polarity of excitability changes vary accordingly to current flow direction, current intensity and stimulation duration [45]. Neuronal excitability changes induced by tDCS are supposed to be mediated by processes which are similar to long-term potentiation (LTP) and depression (LTD), among others [45]. These neurobiological mechanisms could significantly favor the intermodal recalibration induced by our initial protocol [35,46]. Interestingly, neurobiological post-effects can be observed during minutes and up to several hours after the end of tDCS stimulation [47,48]. Lasting clinical benefits on USN were also shown [28,31] but not consistently, due in part to substantial protocol differences (stimulation duration and repetition, concurrent activity, time since brain lesion). This technique has already been used simultaneously to occupational therapy sessions with positive and lasting results [28]. Moreover, its association with “bottom–up” rehabilitation approaches is very promising in terms of potentiating their beneficial impact on USN [32,33]. Associating tDCS with our initial protocol therefore seems perfectly suited to attempt to induce or reinforce LTP-like activity-dependent plasticity and consequently lasting intermodal recalibration to improve clinical aspects of USN.

There are some inherent features of our study protocol that deserve cautious consideration. Particular attention will be directed to the phenomena of limb activation and motor extinction [36,37], since our protocol is designed around the performance of voluntary upper limb movements simultaneously to stimulation sessions in USN patients. Although it has been shown in the literature that the use of the contralesional arm in the neglected hemi-space significantly reduces the severity of USN [37], these patients generally have great difficulty in mobilizing it in the early post-stroke stages. Moreover, the simultaneous use of both upper limbs in the neglected hemi-space limits or even suppresses the previously observed beneficial effects [36,37] possibly by perceptual “overshadowing” of the contralesional arm movements by the ipsilesional ones, according to Robertson and North [37]. Therefore, the therapists will carefully avoid bimanual activities in the neglected hemi-space, but exercises will be allowed with the right upper limb instead of the left one if the latter is too impaired to effectively perform them in initial phases of rehabilitation. This gradual increase in task complexity is also intended to maintain high motivation levels. Other vigilance points will be taken into account. We will be watchful of population homogeneity, hence the need to include a sufficient number of patients. We will compare the evolution of each patient from baseline according to his affiliation group, given the possibility of inter-subject variability in responsiveness. In addition, the long-term effects of tDCS should be clarified, and its optimal targets for motor and spatial rehabilitation have not been sufficiently studied yet. There is also a risk of misuse or overuse of non-invasive stimulations. Finally, local rehabilitation teams will be coordinated for inter-site standardization of inclusion, occupational therapy tasks, test performance and stimulation tools use.

In conclusion, the originality of our study lies in this novel combination of non-invasive stimulation techniques with low attentional demands and their application during occupational therapy sessions, thus ideally complementing conventional rehabilitation. The results of our multimodal approach should also provide interesting data to reconsider hemi-spatial theories suggesting a dominant function of the right hemisphere or others supporting an “interhemispheric competition” model [49].

## Figures and Tables

**Figure 1 brainsci-13-00678-f001:**
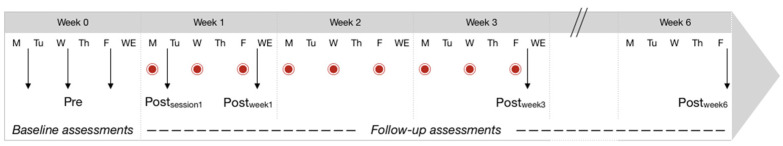
HEMISTIM protocol design. Red circle symbols indicate experimental sessions. Assessment periods are indicated by vertical arrows.

**Figure 2 brainsci-13-00678-f002:**
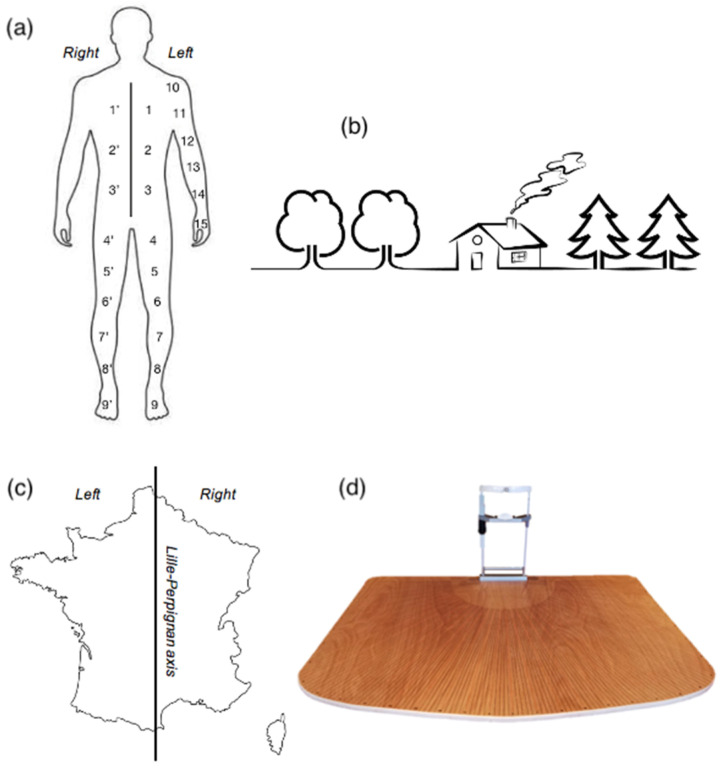
(**a**) Topographical displacement of the stickers in the Fluff test. Numbers indicate target locations; (**b**) Gainotti’s drawing copying test; (**c**) Spatial benchmarks of the map of France; (**d**) Experimental device for the Subjective Straight-Ahead test.

**Figure 3 brainsci-13-00678-f003:**
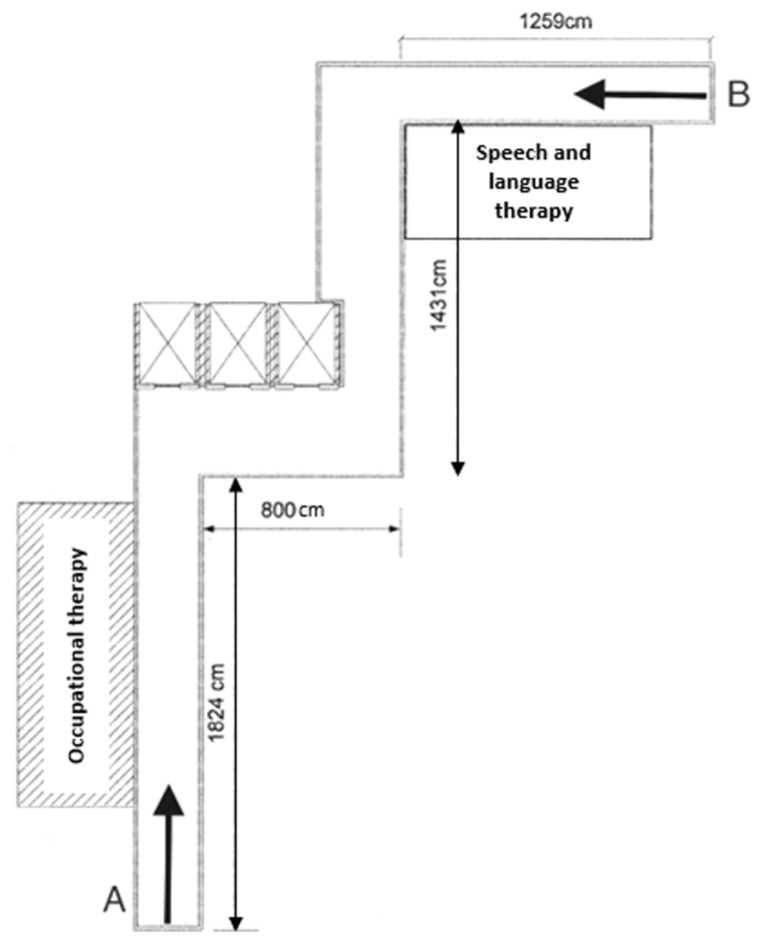
Example of wheelchair navigation course taken from Ceyte et al. (2019) [21].

## Data Availability

No new data were created or analyzed in this study. Data sharing is not applicable to this article.

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
