# Peer review of "Innovative Therapy Combining Neck Muscle Vibration and Transcranial Direct Current Stimulation in Association with Conventional Rehabilitation in Left Unilateral Spatial Neglect Patients: HEMISTIM Protocol for a Randomized Controlled Trial"

_brainsci, 2023, doi:10.3390/brainsci13040678_

Round 1

Reviewer 1 Report

I think this is an exceptional potential RCT and throughly support the publication of protocols papers such as this. 

Author Response

REVIEWER#1' COMMENTS:

I think this is an exceptional potential RCT and throughly support the publication of protocols papers such as this. 

Author’s response: Thank you for your review and support.

Reviewer 2 Report

The authors proposed a protocol (HEMISTIM) to assess immediate and long-term recovery induced by left-side neck-muscle vibration (NMV) and anodal transcranial direct current stimulation (tDCS) on the ipsilesional posterior parietal cortex during occupational therapy sessions in patients with left unilateral spatial neglect (USN). They aimed at recruiting subjects randomly assigned to 4 groups: control, left-NMV, left-NMV + sham-tDCS, and left-NMV + anodal-tDCS. They stated that NMV and tDCS will be applied during the first 15 minutes of occupational therapy sessions, three days a week for three weeks. USN will be assessed at baseline, just at the end of the first experimental session, after the first and third weeks of the protocol, and three weeks after applying treatment. They will evaluate the functional Catherine Bergego Scale score and five tests to investigate the neuropsychological aspects of the USN. They expected that the left NMV would enhance the effects. Moreover, they expected to reinforce lasting intermodal recalibration through LTP-like plasticity induced by anodal tDCS.

 Moreover, the topic is made interesting by combining tDCS with the other techniques during a task that can add important information to the literature.

However, there are some other works showing that cathodal tDCS (c-tDCS) combined with visual feedback improves human balance control (Emadi Andani et al., 2020). It can also be added to support the main idea of the authors, i.e., the effect of tDCS combined with the other techniques during a task to improve brain plasticity, as the authors of this work expected. Conversely, in that work, the c-tDCS was effective, but in that study, they applied it to the cerebellum. The authors should explain clearly why they considered just a-tDCS or add another group with left-NMV + a-tDCS.

Emadi Andani, M., Villa-Sánchez, B., Raneri, F. et al. Cathodal Cerebellar tDCS Combined with Visual Feedback Improves Balance Control. Cerebellum 19, 812–823 (2020). https://doi.org/10.1007/s12311-020-01172-0   

Another issue is computing the sample size. I suggest using G-Power or other software to calculate the sample size. With the 4 groups and 5 sessions, the G-Power also suggests recruiting 48 participants, as the authors mentioned as well. It can be added as support for the sample size calculation. However, with the 5 groups, by adding a-tDCS, the total sample size will be 50, which is not a huge difference.

Faul, F., Erdfelder, E., Lang, A., & Buchner, A. (2007). G*Power 3: A flexible statistical power analysis program for the social, behavioral, and biomedical sciences. Behavior Research Methods, 39(2), 175–191. https://doi.org/10.3758/bf03193146

Author Response

REVIEWER#2' COMMENTS:

The authors proposed a protocol (HEMISTIM) to assess immediate and long-term recovery induced by left-side neck-muscle vibration (NMV) and anodal transcranial direct current stimulation (tDCS) on the ipsilesional posterior parietal cortex during occupational therapy sessions in patients with left unilateral spatial neglect (USN). They aimed at recruiting subjects randomly assigned to 4 groups: control, left-NMV, left-NMV + sham-tDCS, and left-NMV + anodal-tDCS. They stated that NMV and tDCS will be applied during the first 15 minutes of occupational therapy sessions, three days a week for three weeks. USN will be assessed at baseline, just at the end of the first experimental session, after the first and third weeks of the protocol, and three weeks after applying treatment. They will evaluate the functional Catherine Bergego Scale score and five tests to investigate the neuropsychological aspects of the USN. They expected that the left NMV would enhance the effects. Moreover, they expected to reinforce lasting intermodal recalibration through LTP-like plasticity induced by anodal tDCS.

Moreover, the topic is made interesting by combining tDCS with the other techniques during a task that can add important information to the literature. 

Author’s response: Thank you for your insightful review.

However, there are some other works showing that cathodal tDCS (c-tDCS) combined with visual feedback improves human balance control (Emadi Andani et al., 2020). It can also be added to support the main idea of the authors, i.e., the effect of tDCS combined with the other techniques during a task to improve brain plasticity, as the authors of this work expected. Conversely, in that work, the c-tDCS was effective, but in that study, they applied it to the cerebellum. The authors should explain clearly why they considered just a-tDCS or add another group with left-NMV + a-tDCS.

Emadi Andani, M., Villa-Sánchez, B., Raneri, F. et al. Cathodal Cerebellar tDCS Combined with Visual Feedback Improves Balance Control. Cerebellum 19, 812–823 (2020). https://doi.org/10.1007/s12311-020-01172-0   

Author’s response: We thank the reviewer for his comments. Our hypothesis is indeed that the combination of tDCS with other bottom-up rehabilitation strategies might potentiate their effects. This citation was added page 9, line 364. tDCS applied over the cerebellum has not been studied yet in spatial representation issues, but only over the parietal or motor cortices. This suggestion is interesting for future studies, as well as addressing the relationship between spatial perceptions and balance control. We considered only a-tDCS, and not c-tDCS, based on Làdavas et al.’s work, in which cathodal tDCS over the unlesioned PPC canceled beneficial effects of prism adaptation: Làdavas E, Giulietti S, Avenanti A, Bertini C, Lorenzini E, Quinquinio C, et al. a-tDCS on the ipsilesional parietal cortex boosts the effects of prism adaptation treatment in neglect. Restor Neurol Neurosci. 2015;33:647–62.

Another issue is computing the sample size. I suggest using G-Power or other software to calculate the sample size. With the 4 groups and 5 sessions, the G-Power also suggests recruiting 48 participants, as the authors mentioned as well. It can be added as support for the sample size calculation. However, with the 5 groups, by adding a-tDCS, the total sample size will be 50, which is not a huge difference.

Faul, F., Erdfelder, E., Lang, A., & Buchner, A. (2007). G*Power 3: A flexible statistical power analysis program for the social, behavioral, and biomedical sciences. Behavior Research Methods, 39(2), 175–191. https://doi.org/10.3758/bf03193146

Author’s response: In order to perform an a priori calculation with G-Power, we would have to rely on articles that used the same primary endpoint (Catherine Bergego Scale evolution). We are not aware of any study that used this primary endpoint in evaluating NMV or tDCS, and therefore could not estimate precisely the effect size. For that reason, we aimed at getting close to the number of subjects included in studies that had similar stimulation paradigms in order to propose a coherent and feasible number of subjects.

Reviewer 3 Report

This is a very interesting protocol for an important topic. I have some comments and suggestions which may improve the quality of this protocol.

1. The Introduction is very long. It appears that many parts (i.e. electrode configuration) might better fit in the Methods section.

2. There is no justification of the stimulation parameters.

3. Subject should be asked about their expectations.

4. Can subjects identify which treatment (tDCS or SHAM) they received?

5. This reviewer is convinced that males and females respond differently to tDCS (e.g., hormones, skull thickness, neurotransmitters). This should be addressed in the protocol.

6. The limitations of tDCS should be addressed.

7. Responders and non-responders should be identified first.

Author Response

REVIEWER#3' COMMENTS:

This is a very interesting protocol for an important topic. I have some comments and suggestions which may improve the quality of this protocol.

Author’s response: Thank you for your review and relevant comments.

  1. The Introduction is very long. It appears that many parts (i.e. electrode configuration) might better fit in the Methods section.

Author’s response: Considering the nature of our original article (study protocol), we voluntarily emphasized the scientific context in the introduction in order to rationalize our methodological choices.

  1. There is no justification of the stimulation parameters.

Author’s response: Stimulation parameters were chosen according to our previous research with NMV only (mentioned page 4, lines 163-165) and using tDCS (page 4, lines 168-171).

  1. Subject should be asked about their expectations.

Author’s response: Thank you for this suggestion. We will question them at the end of each week of intervention (specified page 7, lines 276-268).

  1. Can subjects identify which treatment (tDCS or SHAM) they received?

Author’s response: Patients will not be able to identify if they received active or sham tDCS, thanks to the reproduction of the initial itching sensation at the beginning of active tDCS in the sham condition. This information is specified page 4, lines 191-194.

  1. This reviewer is convinced that males and females respond differently to tDCS (e.g., hormones, skull thickness, neurotransmitters). This should be addressed in the protocol.

Author’s response: As specified page 8, lines 289, patients will indeed be randomized and comparability between groups regarding sex (but also other parameters such as age, stroke volume, severity of deficit, etc.) will be checked before further analysis. Thus, even if males and females may respond differently, this difference would be non-differential between groups. However, differences in responsiveness will be considered, specified page 9, lines 391-392.

  1. The limitations of tDCS should be addressed.

Author’s response: Limitations of tDCS were added pages 9-10, lines 392-395.

  1. Responders and non-responders should be identified first.

Author’s response: We did not plan to identify responders and non-responders a priori because the expected effect of our interventions is medium and long term. We agree that inter-subject variability in effectiveness is possible and expect to be able to identify a posteriori which subjects respond better than others. This is specified in the limitations page 9, lines 389-392.